# Comparison of Different Isolation Methods for Plasma-Derived Extracellular Vesicles in Patients with Hyperlipidemia

**DOI:** 10.3390/life12111942

**Published:** 2022-11-21

**Authors:** Ke Zhen, Xiaojuan Wei, Zelun Zhi, Shuyan Zhang, Liujuan Cui, Yue Li, Xia Chen, Jing Yao, Hongchao Zhang

**Affiliations:** 1The First Clinical Medical College, Hebei North University, Zhangjiakou 075132, China; 2Department of Cardiovascular Surgery, Air Force Medical Center, People’s Liberation Army, Beijing 100048, China; 3Institute of Biophysics, Chinese Academy of Sciences, Beijing 100101, China

**Keywords:** plasma-derived extracellular vesicle, hyperlipidemia, ultracentrifugation, polyethylene glycol precipitation, size exclusion chromatography

## Abstract

Extracellular vesicles are commonly found in human body fluids and can reflect current physiological conditions of human body and act as biomarkers of disease. The quality of isolated extracellular vesicles facilitates the early diagnosis of various diseases accompanied by hyperlipidemia. Nonetheless, there are no reports on which special methods are suitable for isolating extracellular vesicles from the plasma of patients with hyperlipidemia. Thus, this study compared three different research-based extracellular vesicle isolation approaches, namely ultracentrifugation (UC), polyethylene glycol (PEG) precipitation, and size exclusion chromatography (SEC), and determined which of them was the most effective method. We selected blood samples from 12 patients with clinically diagnosed hyperlipidemia and isolated plasma-derived extracellular vesicles using three methods. The morphology of the isolated extracellular vesicles was observed using transmission electron microscopy, while the concentration was detected by asymmetric flow field-flow fractionation and multi-angle light scattering. Marker proteins were identified by Western blotting, and protein composition was evaluated by silver staining. Both determined the contaminations in the extracellular vesicle samples. The results showed that the three methods can be successfully used for the isolation of extracellular vesicles. The extracellular vesicles isolated by UC were larger in size, and the yield was much lower. Although the yield of extracellular vesicles isolated by PEG precipitation was greatly improved, the contamination was increased. Of the three methods, only the SEC-isolated extracellular vesicles were characterized by high yield and low contamination. Therefore, our data suggested that the SEC was a more ideal method for isolating extracellular vesicles from the plasma of patients with hyperlipidemia.

## 1. Introduction

Extracellular vesicles (EVs), including exosomes, microvesicles, and apoptotic bodies, are nanoscale vesicles secreted by cells. They have a bilayer phospholipid membrane structure carrying complex substances, including nucleic acids, proteins, and lipids [1]. EVs play an important role in intercellular communication [2,3]. Recently, EVs have received widespread attention in diagnosis and therapy as disease biomarkers and targeted drug carriers [4,5,6,7,8,9,10,11,12]. However, the quality of isolated EVs significantly affects the results of related studies, limiting their clinical applications [13]. Precise isolation of EVs is thus one of the key methodological underpinnings of downstream research. Notably, the fact that EVs may derive from different sources makes it difficult to select an ideal isolation method that is simultaneously efficient, high-quality, and reproducible [14]. Until now, there has been no standard method available for EV isolation for each sample type. The plasma sample of hyperlipidemia is complex, and the number of plasma proteins, lipoprotein particles, and cytokines secreted by tissues or organs is large, which increases the difficulty of EV isolation [15]. Recently, many studies have reported state-of-the-art methods to isolate, separate, identify, and quantify EVs from plasma [16,17,18,19,20]. The interactions between EVs and related antibodies are also analyzed using precise electronic devices such as quartz crystal microbalances [21]. However, an appropriate approach to isolating EVs from plasma is urgently needed to acquire high-quality EVs and boost further research on them.

Ultracentrifugation (UC) is a traditional isolation method for isolating EVs from human body fluids [22]. Apart from UC, EVs can also be isolated using ultrafiltration, size exclusion chromatography (SEC), polyethylene glycol (PEG) precipitation, the immunomagnetic bead method, asymmetric flow field-flow fractionation (AF4), and commercial EV isolation kits [15,23,24,25,26]. However, owing to various limitations, such as the easy clogging of ultrafiltration membranes, the low isolation efficiency of the immunomagnetic bead method, and the need for specialized collection equipment for AF4, most of these methods have not been widely used [1]. PEG precipitation and SEC represent the separation principles of most of the currently available EV isolation kits [14]. PEG precipitates EVs through gravitational depletion and volume exclusion [27]. SEC separates EVs depending on the vesicle size and has been reported to be the best method for isolating EVs [14].

In clinical research, blood samples are frequently used. Numerous studies have utilized bicinchoninic acid (BCA) to determine the concentration of EV proteins; Western blotting (WB) to identify the EV-specific marker proteins; and CD9, CD63, and HSP70 to evaluate the quality of isolated EVs [28]. However, current data from databases, including the ExoCarta, exoRBase, and EV-TRACK, have not shown whether EVs yield a stable expression of housekeeping genes, making it difficult to judge the quality of the isolation by comparing the protein expression of EVs. The composition of plasma is also quite complex, and plasma-derived EVs are often contaminated with plasma proteins, including albumin and globulin or lipoproteins (e.g., high-density lipoprotein (HDL), low-density lipoprotein (LDL), very low-density lipoprotein (VLDL), LP(a), and chylomicron (CM)) [29]. Therefore, it is ambiguous to compare only the protein amounts or particle sizes of different isolation methods for plasma-derived EVs, especially in patients with hyperlipidemia.

In this study, we analyzed three methods (UC, PEG precipitation, and SEC) for isolating EVs from the plasma of patients with hyperlipidemia. Exploring a more appropriate method of isolating EVs from the plasma in these patients promises to provide new insights into the study of EVs.

## 2. Materials and Methods

### 2.1. Sample Collection

Twelve patients (Table 1) with different types of hyperlipidemias treated in the Department of Cardiovascular Surgery, Airforce Medical Center, were included in this study. This study was approved by the Ethics Committee of the Airforce Medical Center and conducted in accordance with the Declaration of Helsinki. All patients provided written informed consent. Before blood collection, the patients had been instructed to be on a light diet for more than 3 days and to fast for 12 h. The whole blood was collected in the early morning of the following day with a vacuum blood collection containing EDTA and stored at 4 °C. Plasma isolation was completed within 1 h. Briefly, the supernatant was isolated from fresh whole blood after centrifugation at 1600× *g* for 10 min at 4 °C, followed by another centrifugation at 16,000× *g* for 30 min at 4 °C, which was a critical step to remove platelet contamination. Finally, the separated supernatant was filtered through 0.22 μm filters (Millipore) and dispensed in 1 mL quantities into Eppendorf tubes which were stored at −80 °C for EV isolation.

### 2.2. EV Isolation and Identification

#### 2.2.1. UC 

UC was performed using the method previously described with minor modifications [30]. Briefly, 3 mL plasma was diluted with 9 mL PBS in an SW40 ultracentrifuge tube and centrifugated at 110,000× *g* for 2 h at 4 °C. The supernatant was discarded, and the pellet was resuspended with 1 mL PBS, diluted with another 9 mL PBS, and centrifugated at 110,000× *g* for 70 min at 4 °C (Figure 1). The procedure was repeated twice, and finally, the pellet was resuspended with 100 µL PBS and stored at −80 °C. All procedures were performed at 4 °C or on ice.

#### 2.2.2. PEG Precipitation

A plasma-derived EV purification kit (Umibio Science and Technology, Shanghai, China) was used to isolate 1 mL plasma-derived EVs for PEG precipitation. Plasma, PBS, and Blood PureExo Solution were added to an Eppendorf tube in a volume ratio of 1:3:1. The mixtures were vortexed for 1 min and then incubated for 2 h at 4 °C. Then the sample was centrifuged at 10,000× *g* for 60 min at 4 °C, and the supernatant was discarded. The pellet was resuspended with 500 µL of PBS per mL of plasma and centrifuged at 12,000× *g* for 2 min at 4 °C. Finally, the supernatant was added to the EV purification filter and centrifuged at 3000× *g* for 10 min at 4 °C (Figure 1).

#### 2.2.3. SEC

An exclusion column (ABclonal Biotechnology, Wuhan, China) for isolation of the EVs was equilibrated at 37 °C for 30 min and used after rinsing with 20 mL PBS. The top space of the column was filled with 1 mL plasma. After the plasma had passed through the sieve plate, 1.5 mL PBS was added for elution. Each 500 µL of effluent was defined as a fraction, and the first three fractions were discarded. Fractions 4–8 enriched in EV particles were collected immediately thereafter, meaning that 2.5 mL of flow-through fluid needed to be collected. The EVs were concentrated with or without a 100 kDa ultrafiltration tube and centrifuged at 4000× *g* for 10 min at 4 °C (Figure 1).

### 2.3. Cell Culture and EV Internalization

The human cardiomyocyte cell line AC16 was a gift from Dr. Shenglan Gao (Department of Cell and Genetic Medicine, School of Basic Medical Sciences, Fudan University). AC16 was cultured in high-glucose Dulbecco’s modified Eagle’s medium (DMEM) containing 10% FBS and 1% penicillin/streptomycin at 37 °C in 5% CO_2_. When the cell density reached approximately 80%, the cells were subcultured in a 1:2 ratio. AC16 was seeded in 24-well plates with coverslips. EVs were labeled with PKH67 according to the manufacturer’s protocol. Briefly, 25 µL of PKH67 working solution was added per 100 µg of EVs, vortexed for 1 min, and then left standing for 10 min. After the cells were plastered, the medium was changed to FBS-free medium, and then PKH67-labeled EVs were added. The mitochondria were stained with MitoTracker Red (Invitrogen, M46752, Carlsbad, CA, USA), while the nuclei were stained with Hoechst 33342 (Beyotime, C1022, Beijing, China) in living cells. The EVs were co-incubated with cells for 24 h at 37 °C in 5% CO_2_, and the free EVs were washed away with PBS. The localization of the EVs in cells was observed under a confocal microscope. The PKH67-labeled EVs were also placed on slides for microscopic observation alone. The internalization of the EVs was analyzed by ImageJ (version 2.0.0).

### 2.4. Silver Staining

The EV samples were lysed directly with 2× sodium dodecyl sulfate (SDS) sample buffer after the indicated treatments, sonicated twice (9 s/time at 30 W), and denatured at 95 °C for 5 min. Then, 2 µL of the prepared EV sample was added to the wells of the 12% polyacrylamide gel. After the end of the polyacrylamide gel electrophoresis, the gel was fixed in fixation buffer (40 mL ethyl alcohol, 10 mL acetic acid, and 40 mL ddH_2_O) for 30 min and then incubated in sensitization buffer (30 mL ethyl alcohol, 0.314 g sodium thiosulfate pentahydrate, 11.28 g sodium acetate trihydrate, and 70 mL ddH_2_O) for 30 min at 25 °C. The gel was washed four times with double-distilled H_2_O for 5 min each time. The washed gel was treated with silver staining buffer (0.25 g silver nitrate, 100 mL ddH_2_O, and 40 µL formaldehyde) for 20 min at 25 °C, followed by developing buffer (5 g sodium carbonate, 100 mL ddH_2_O, and 40 μL formaldehyde) until the bands appeared. The reaction was blocked in stopping buffer (1.68 g ethylenediaminetetraacetic acid disodium salt and 100 mL ddH_2_O) immediately after achieving a satisfactory resolution.

### 2.5. Western Blotting

After the total protein loading of each EV sample was made consistent by silver staining, the total proteins were then separated by SDS-PAGE; transferred to PVDF membranes; determined with antibodies, including CD63 (Cat#A19023, ABclonal, Wuhan, China), TSG101 (Cat#A5789, ABclonal, Wuhan, China), HSP70 (Cat#A0284, ABclonal, Wuhan, China), calnexin (Cat#A4846, ABclonal, Wuhan, China), ApoA1 (Cat#A4163, ABclonal, Wuhan, China), ApoA4 (Cat#5700S, Cell Signaling Technology, Danvers, MA, USA), ApoE (Cat#A0304, ABclonal, Wuhan, China), ApoB100 (Cat#A4184, ABclonal, Wuhan, China), albumin (Cat#A11625, ABclonal, Wuhan, China), human IgG (Cat#A19711, ABclonal, Wuhan, China), and human IgM (Cat#A19719, ABclonal, Wuhan, China); and detected using the ECL system (LABLEAD, Beijing, China).

### 2.6. AF4 Coupled with Multi-Angle Light Scattering (AF4-MALS)

The isolated EV fractions were diluted with PBS (pH 7.2–7.4, Cat#P1020, Solarbio Life Science, Beijing, China, filtered through a 0.22 μm membrane) to obtain a protein concentration of around 1 mg/mL to prevent clogging of the field flow membrane (10 kDa RC, ROAB08766, Wyatt Technology, Santa Barbara, CA, USA). The overall AF4-MALS process was conducted at 25 °C. The relevant parameters were set to 1.00 mL/min for detector flow, 1.50 mL/min for focus, and 0.20 mL/min for injection [25,31]. Fifty-microliter EV samples were injected for each assay, and the conditions are shown in Appendix A. The geometric radius of the EVs was determined based on the MALS results using a sphere model, and the particle density was calculated using the Astra software (version 5.4.4.20, Wyatt Technology).

### 2.7. DLS

Dynamic light scattering (DLS) was performed using the Delsa Nano C Particle Analyzer (Beckman Coulter, Indianapolis, IN, USA). The EV suspensions were measured using a 3.5 mL quartz cuvette at 25 °C. The data were analyzed using the Delsa Nano C software Version 2.31.

### 2.8. TEM

The purified EVs were loaded on a 200-mesh formvar-coated copper grid for 1 min, and the excess solution was removed using filter paper. Thereafter, the grid was stained with 1% (*w*/*v*) uranyl acetate for 1 min, and the excess solution was removed using filter paper. The grid was observed under a Tecnai Spirit electron microscope (FEI, Eindhoven, The Netherlands).

### 2.9. Statistical Analysis

All statistical calculations were performed using GraphPad Prism software (version 9.0). The results were presented as means ± SDs. Statistical analysis between two groups was performed using an unpaired Student’s *t*-test, and *p* < 0.05 was considered statistically significant.

## 3. Results

### 3.1. The Size of the EVs Isolated by UC Is Larger Than That by PEG Precipitation and SEC

Morphology is one of the main features of EVs required by ISEV [32]. In addition, we can characterize EVs by particle size analysis and EV-specific marker proteins together. In this study, TEM showed that EVs isolated by all three methods presented a typical bilayer membrane structure (Figure 2A). In addition, the EVs isolated by UC had a larger diameter, intact structure, and less contamination (Figure 2A,B). The EVs isolated by PEG precipitation were relatively uniform in size distribution and had an intact structure, but there were massive contaminations of PEG polymers and other particles (Figure 2A). Meanwhile, the EVs isolated by SEC were intact and had a small number of other particles (Figure 2A). Similar to the DLS results, the average size of the UC-isolated EVs (253.02 ± 94.15 nm) was significantly larger than that of the EVs isolated by PEG precipitation (50.37 ± 14.16 nm, *p* < 0.001) and SEC-isolated EVs (49.90 ± 3.74 nm, *p* < 0.001) (Figure 2B,C). In addition, the size distribution of the EVs isolated by SEC was more monodisperse than that of the EVs isolated by UC and PEG precipitation.

### 3.2. The Proportion of the EVs Isolated by UC Is Smaller Than That by PEG Precipitation and SEC

Exosomes are a major subtype of EVs sized 30 to 150 nm. The proportion of the EVs isolated by UC in the range of 30–150 nm was relatively lowest at 45.08% ± 10.22% compared with that of EVs isolated by PEG precipitation (91.03% ± 9.07%, *p* < 0.0001) and SEC (96.03% ± 4.46%, *p* < 0.0001) (Figure 2D and Appendix A). The size distributions of EVs were represented by cumulative size distribution values D10, D50, and D90 (Figure 2E). The vesicle sizes isolated by PEG precipitation and SEC mostly corresponded to the vesicle size of the EVs, while approximately 50% of the vesicles isolated by UC were outside the vesicle size of the EVs.

### 3.3. The EVs Isolated by SEC Showed a High Yield and Low Contamination

The EVs were labeled with PKH67 and observed under a confocal laser scanning microscope. Similar to the TEM results, aggregation was observed in the EVs isolated by PEG precipitation (Figure 3A). To further characterize the EVs, we identified the expression of positive EV marker proteins TSG101, CD63, and HSP70 and negative marker protein calnexin in the three methods using WB (Figure 3B). The results showed that all samples contained EV-specific marker proteins, suggesting that it was feasible to isolate plasma-derived EVs from patients with hyperlipidemia using three methods. The highest content of EV-specific marker proteins was isolated using PEG precipitation, the lowest content of EV-specific marker proteins was isolated using UC, and the content of EV-specific marker proteins isolated using SEC fell between those of PEG and UC. These findings indicated that the yield of EVs increases sequentially from PEG precipitation to SEC and UC. Lipoprotein particles similar in size to EVs or proteins with higher abundance in plasma are likely to be obtained together during EV isolation. To identify the extent of contamination of the plasma proteins and lipoprotein particles in the EV samples, we measured the levels of albumin, IgM, IgG, Apo-A1, Apo-A4, Apo-E, and Apo-B100 by WB (Figure 3C). The protein expression of Apo-A1, Apo-A4, and Apo-E was significantly higher in the EVs isolated by PEG precipitation than in EVs isolated by UC and SEC, which meant that HDL was mainly present in EVs isolated by PEG precipitation, followed by VLDL and LDL (Figure 3C), whereas the EVs isolated by SEC are mainly VLDL, LDL, and some HDL. In general, all three methods preserved more or less unnecessary protein contamination. However, the PEG precipitation-isolated EVs retained the most contamination, followed by the SEC- and UC-isolated EVs. AF4-MALS was used to sort the EVs by diameter and determine the number of proteins carried by the EVs of different sizes. We found that UC and PEG precipitation isolated larger amounts of proteins from larger EVs, while SEC isolated larger amounts of proteins from smaller EVs (Figure 4A–C). The EV concentration was defined as the number of EVs included in 1 mL of plasma. The concentrations of the EVs isolated by PEG precipitation and SEC were similar, and both were higher than those isolated by UC (Figure 4D). The number of proteins in EVs can also reflect the concentration of EVs to some extent. In the BCA protein analysis, the protein concentrations of the EVs isolated by UC, PEG precipitation, and SEC were 0.11 ± 0.04, 24.14 ± 3.90, and 1.14 ± 0.17 μg/μL, respectively (Figure 4E). The protein concentration of the EVs isolated by PEG precipitation was significantly higher than that of the SEC and UC methods (*p* < 0.0001). Meanwhile, the protein concentration of the EVs isolated by SEC was higher than that of EVs isolated by UC (*p* < 0.0001). To verify the biological activity of the EVs isolated using the three methods, we co-incubated the PKH67-labeled EVs with AC16 cells for 24 h. After washing the cells sufficiently with PBS, the EVs were observed under an Olympus FV3000 confocal microscope to see if they could be incorporated into the cells. The analysis showed that the EVs isolated by the three methods could be incorporated into cells, and there was no significant difference in the internalization of the EVs isolated by UC (*p* < 0.001) and SEC (*p* < 0.001) (Figure 4F,G). However, under the same conditions, the internalization of the EVs isolated by PEG precipitation was significantly greater than that of EVs isolated by UC and SEC. Notably, if PEG precipitation-isolated EVs are used in EV internalization experiments, it may not be possible to distinguish whether the cells are incorporating EVs or lipoprotein particles. The internalization of the EVs was quantified using the ImageJ software (version 2.0.0).

## 4. Discussion

Since Johnstone discovered EVs in sheep erythrocyte supernatants in 1987, there has been no ideal and uniform method for isolating EVs from a variety of samples [2,33,34]. Special samples require a comparison of isolation methods. Thus, researchers have been attempting to explore a suitable method of EV isolation for each sample. Hyperlipidemia is a common predisposing factor for CVD [35,36]. To our knowledge, no methodology has been studied to investigate the isolation of EVs from the plasma of patients with hyperlipidemia.

EVs act as cross-talk mediators, regulating the function of different cells and organs [37,38]. In the past decades, EVs had been isolated by UC [39]. However, the practical application of UC is greatly limited due to the high demand for sample quantity, long procedural duration, and expensive equipment [33]. In our study, we found that the amount of the EV proteins isolated by UC was much lower than that with the other methods, despite the sample volume used for PEG precipitation and SEC being 3 times greater (Figure 4E). Rider et al. found that PEG precipitation can be used not only to purify viruses, but also to rapidly isolate EVs from cell culture media in large quantities [40]. It represents a new method for EV isolation and effectively compensates for the low yield of UC. Owing to the low cost and high yield of PEG precipitation for EV isolation, PEG precipitation has become the most popular method after UC [41,42]. In this study, we also used PEG precipitation and obtained similar results: a larger number of EVs per milliliter of plasma was isolated by PEG precipitation than by UC. We also found that the EVs isolated by PEG precipitation had a much smaller particle size than those isolated by UC [14]. The centrifugal force of UC is greater than that of PEG precipitation, resulting in the fusion of the vesicles after fragmentation, which may explain the large particle size of the UC-isolated EVs. Plasma-derived EVs are thought to be derived from multiple cell populations, and their heterogeneous origin may affect the detection of disease-specific biomarkers. In addition, the presence of a large number of lipoprotein particles in the plasma of patients with hyperlipidemia may also dilute the percentage of EVs. In our study, the PEG precipitation-isolated EVs were contaminated with plasma and lipoprotein particles, which is consistent with previous reports [42,43]. Therefore, PEG precipitation may not be suitable as an EV isolation method for patients with hyperlipidemia.

To ensure the yield of EVs while reducing the contamination of non-EV proteins, we attempted to isolate the EVs using SEC, which is currently considered the best method for isolating and purifying EVs from various biofluids [14,44]. In our study, the yield of the EVs isolated by SEC was approximately 83 times that of EVs isolated by UC, and SEC greatly reduced the number of unnecessary proteins compared to PEG precipitation (Figure 4E). Compared to PEG precipitation and UC, the SEC column can be used repeatedly, which makes it more suitable for clinical samples and yields high-quality and high-purity EVs. Although lipoprotein particles in the EVs isolated by SEC were not completely removed, there was little effect on the physiological activity of the EVs (Figure 3B and Figure 4F). It is worth noting that EVs generally have a bilayer phospholipid membrane structure, and if lipids in the exosomal microenvironment are excessively removed, the homeostasis of the human body environment may be ignored, and it is difficult to accurately reflect the actual effects in vivo [45,46]. Evidently, SEC-isolated EVs are more physiologically compatible. However, their actual state in vitro and in vivo and the effect of lipoproteins on these EVs need to be further studied. In conclusion, SEC is more suitable than PEG precipitation and UC for the isolation of EVs from the plasma in patients with hyperlipidemia in terms of accessibility, yield, and purity (Table 2). These findings can provide a methodological basis for further functional and mechanistic studies of plasma-derived EVs in patients with hyperlipidemia, including the evaluation of the relationship between EVs and CVD or other lipid metabolism diseases.

This study had some limitations. First, our sample size was limited owing to the difficulty of obtaining clinical samples. The results need to be validated in studies with a larger sample size. Second, we were unable to verify the sensitivity and specificity of the three methods based on identified EV disease biomarkers. In future studies, a method that could combine the advantages of various methods and advance the research on EVs should be explored.

## Figures and Tables

**Figure 1 life-12-01942-f001:**
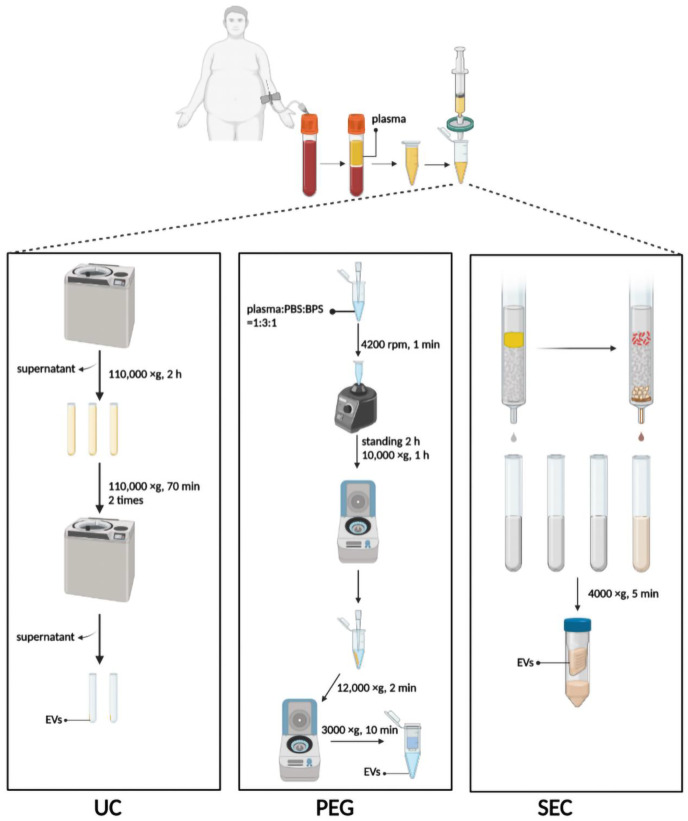
Schematic diagram of three methods to isolate EVs. UC, ultracentrifugation; PEG, polyethylene glycol precipitation; SEC, size exclusion chromatography. Figure created with Biorender.com.

**Figure 2 life-12-01942-f002:**
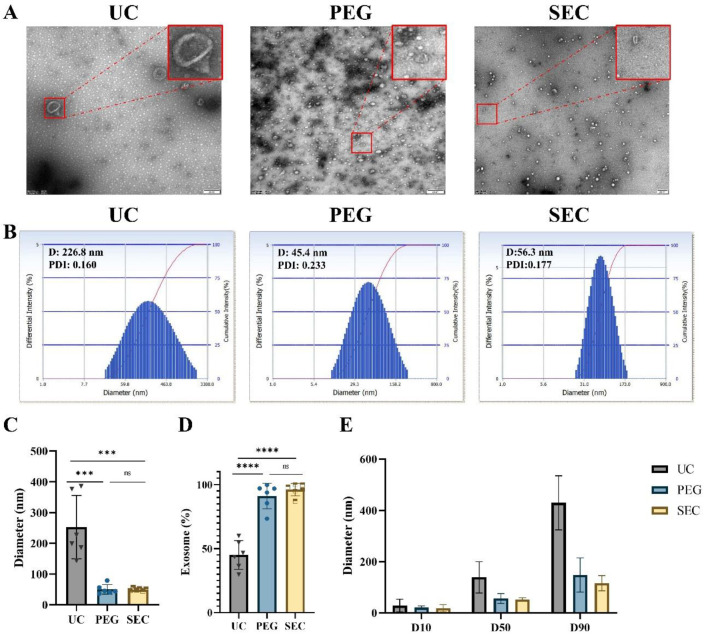
Isolation and characterization of EVs. (**A**) Transmission electron microscopy images of EVs isolated by ultracentrifugation (UC), polyethylene glycol precipitation (PEG), and size exclusion chromatography (SEC), bar = 200 nm. (**B**) The size distributions of EVs isolated by the three methods were determined by DLS. Quantitative analysis of (**C**) mean EV diameter (*n* = 6). (**D**) Percentage of exosomes in EVs (*n* = 6). (**E**) Various particle size distribution diameters were measured (D10, D50, and D90, *n* = 3), means ± SD. The comparisons between samples are indicated by lines, and the statistical significance is indicated by asterisks above the lines. No significance (ns) indicates *p* > 0.05, *** *p* < 0.001, **** *p* < 0.0001.

**Figure 3 life-12-01942-f003:**
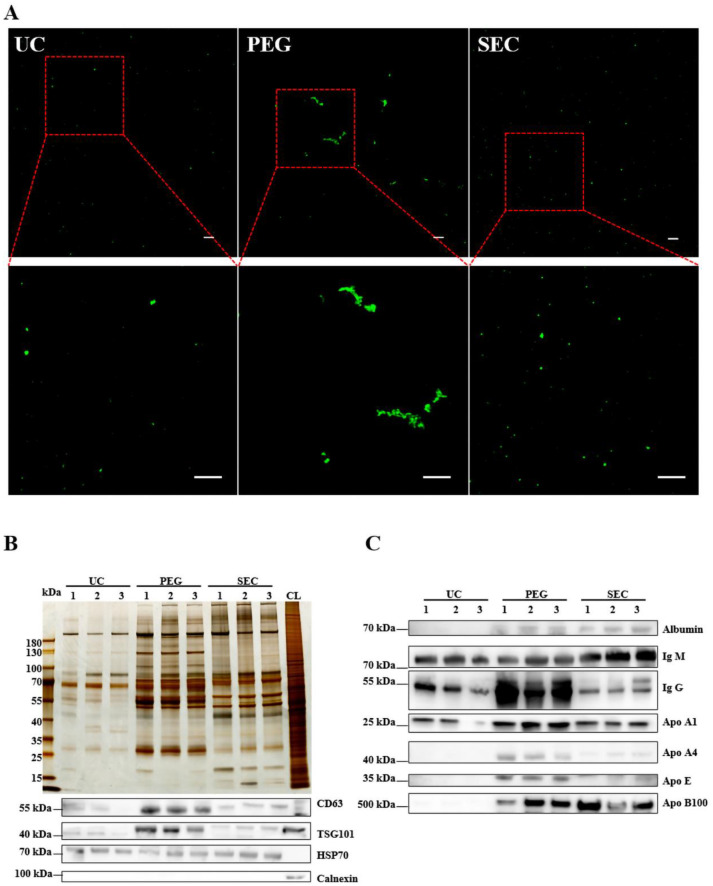
Identification of the quality of EVs isolated by three methods. (**A**) EVs isolated by UC, PEG, and SEC methods were stained with PKH67 (100 μM) and observed with an Olympus FV3000 confocal microscope (*n* = 10). Bar = 5 μm. (**B**) The differential protein concentrations isolated by the three methods were normalized and compared with silver staining, and the differential protein concentrations of EVs CD63, HSP70, TSG101, and calnexin were observed by WB. (**C**) Representative WB images showed the expression of albumin, IgM, IgG, ApoA1, ApoA4, and ApoE.

**Figure 4 life-12-01942-f004:**
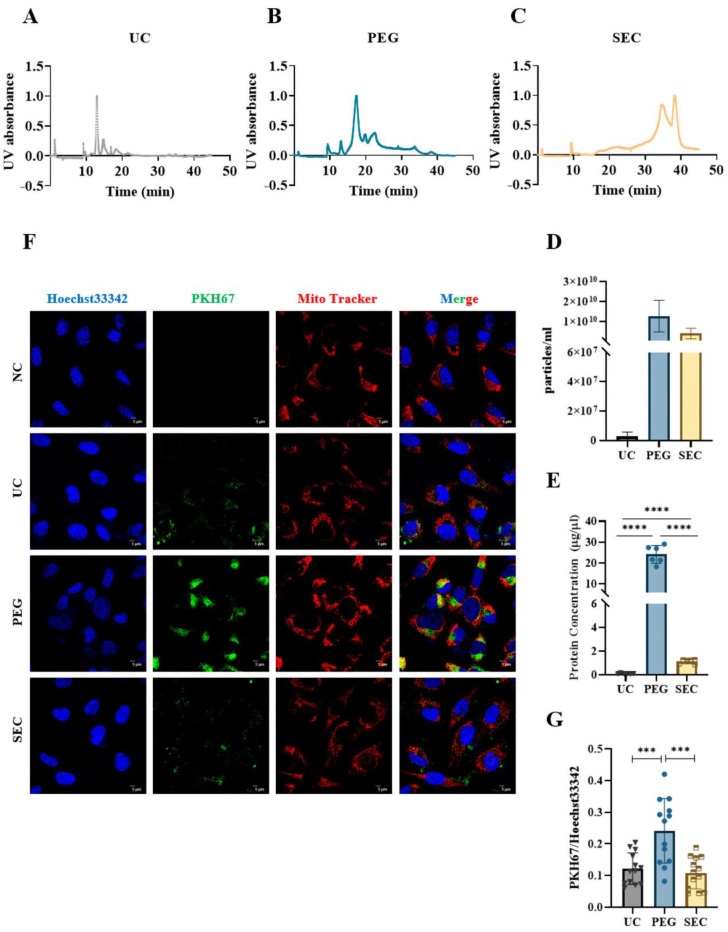
Comparison of the concentration and internalization of EVs isolated by different isolation methods. (**A**–**C**) Absorbance of EVs isolated by UC, PEG, and SEC methods at different times at UV 280 nm. (**D**) EVs isolated from the three samples were analyzed by AF4-MALS (*n* = 3). (**E**) The protein concentrations of EVs collected by three methods were measured by BCA protein assay. (**F**) Representative images of EVs incorporating Mito-Tracker (Red, 1:1000), PKH67 (Green, 100 μM), and Hoechst 33342 (Blue, 10 μg/mL). Bar = 5 μm. (**G**) Quantification of PKH67^+^ staining in (**A**) (*n* = 13). The comparisons among samples are indicated by lines, and the statistical significance is indicated by asterisks above the lines. NS indicates *p* > 0.05, *** *p* < 0.001, **** *p* < 0.0001.

**Table 1 life-12-01942-t001:** Patient characteristics.

Characteristics	
Age, years, mean ± SD	62.42 ± 8.79
Sex (%)	Male (58.33%), Female (41.67%)
Body mass index, kg/m^2^, mean ± SD	27.42 ± 2.19
Obesity (%)	33.33%
Diabetes (%)	41.67%
CVD (%)	100%
THO, mean ± SD	4.84 ± 1.33 mmol/L
TG, mean ± SD	1.90 ± 0.63 mmol/L
HDL, mean ± SD	1.05 ± 0.33 mmol/L
LDL, mean ± SD	3.03 ± 1.13 mmol/L
LP(a), mean ± SD	174.67 ± 175.64 mg/L

CVD: cardiovascular disease, THO (normal range: 2.80–5.18 mmol/L): total cholesterol, TG (normal range: 0.51–0.70 mmol/L): triacylglycerol, HDL (normal range: 1.04–2.00 mmol/L): high-density lipoprotein, LDL (normal range: 2.00–3.37 mmol/L): low-density lipoprotein, LP(a) (normal range: 0–300 mg/L): lipoprotein(a).

**Table 2 life-12-01942-t002:** Characteristics of the EV isolation methods.

Characteristic	UC	PEG Precipitation	SEC
Plasma volume (mL)	3.0	1.0	1.0
Concentration (particles/mL plasma)	8.24 × 10^6^ ± 7.59 × 10^6^	1.27 × 10^10^ ± 6.50 × 10^9^	4.03 × 10^9^ ± 2.20 × 10^9^
Mode (nm)	253.02 ± 94.15	50.37 ± 14.16	49.90 ± 3.74
Yield (30–150 nm particles)	45.08% ± 10.22%	91.03% ± 9.07%	96.03% ± 4.46%
Contamination	Low	High	Low
Time (h)	4.0–5.0	3.0–4.0	1.0–2.0
Operability	Cumbersome	Cumbersome	Convenient
Cost	High	Low	Low

## Data Availability

The study did not report any data.

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
