# Peer review of "Comparison of Different Isolation Methods for Plasma-Derived Extracellular Vesicles in Patients with Hyperlipidemia"

_life, 2022, doi:10.3390/life12111942_

Round 1

Reviewer 1 Report

Authors are using exosome throughout the paper, and define it: Exosomes are a major subtype of extracellular vesicles sized 30 to 150 nm. But for example, UC samples produced in this manuscript are not of this defined size. In addition, according to MISEV there is a need for proof that particles are from the endosomal compartment for them to be called exosomes. This proof is not present. Thus it would be suggested that throughout the text extracellular vesicles would be used instead of exosomes. In those cases where exosome size criteria are met in the results section, authors can use exosome-sized particles. 

line 36. Change exosomes to extracellular vesicles throughout the text

lines 50-60. State of the art of plasma isolation of extracellular vesicles is missing. See following publications:

Multia, Evgen, et al. "Fast isolation of highly specific population of platelet-derived extracellular vesicles from blood plasma by affinity monolithic column, immobilized with anti-human CD61 antibody." Analytica Chimica Acta 1091 (2019): 160-168.

Morani, Marco, et al. "Electrokinetic characterization of extracellular vesicles with capillary electrophoresis: A new tool for their identification and quantification." Analytica Chimica Acta 1128 (2020): 42-51.

Multia, Evgen, et al. "Automated on-line isolation and fractionation system for nanosized biomacromolecules from human plasma." Analytical chemistry 92.19 (2020): 13058-13065.

Liangsupree, Thanaporn, Evgen Multia, and Marja-Liisa Riekkola. "Modern isolation and separation techniques for extracellular vesicles." Journal of Chromatography A 1636 (2021): 461773.

Liangsupree, Thanaporn, et al. "Kinetics and interaction studies of anti-tetraspanin antibodies and ICAM-1 with extracellular vesicle subpopulations using continuous flow quartz crystal microbalance biosensor." Biosensors and Bioelectronics 206 (2022): 114151.

Liangsupree, Thanaporn, et al. "Raman spectroscopy combined with comprehensive gas chromatography for label-free characterization of plasma-derived extracellular vesicle subpopulations." Analytical Biochemistry 647 (2022): 114672.

lines 68-72. Authors specify that LDL and VLDL are contaminants, but do not show presence or lack of LDL and VLDL contaminants in the isolates by measuring Apolipoprotein B-100 (apo B-100) (molecular weight 500 kDa) in the isolates.

lines 87: blood collection containing EDTA - have authors researched the literature and looked how EDTA affect the isolation compared to e.g., citrate?

Table 1: LDL, mean±SD 3.03±1.13 mmol/L is the highest concentration in patient samples, but authors do not show any measurements of Apob-100 in the results.

line 139: three differnet protocols -> different

line 141: Created with Biorender.com: what is created? is this at wrong place, should be with the figure 1?

lines 153-163: Major contaminant in all of the three isolation methods is usually LDL and VLDL. Why authors are not measuring Apolipoprotein B-100 (apo B-100) (molecular weight 500 kDa) in the isolates.

Line 199:the average size of the UC-isolated exosomes (253.02±94.15 nm). Authors define Exosomes as extracellular vesicles sized 30 to 150 nm. According to this criteria UC isolates cannot be called exosomes. 

lines 233-238: Apolipoprotein B-100 (apo B-100) (molecular weight 500 kDa) would be the major contaminant and it is missing. 

lines 237-240:The protein expression of Apo-A1, Apo-A4, and Apo-E was significantly higher in the exosomes isolated by PEG precipitation than in exosomes isolated by UC and SEC, which meant that there was a large amount of HDL in exosomes isolated by PEG precipitation (Fig. 3C). 

apoE and apoA4 are mainly found in VLDL, which is also of more fitful size to the isolates, thus it is not justified to say that the contaminant is mainly HDL. The contaminants are most likely LDL, VLDL and of some part HDL. This should be shown by apoB100 Elisa or western blot of apoB100.

line 345: Table 2: Pollution.

Why this word is used, when throughout the manuscript contamination was used?

Reviewer 2 Report

In this manuscript, the authors compared 3 different isolation methods for plasma-derived exosomes in patients with hyperlipidemia, concluding that SEC was an ideal method for isolating exosomes from the plasma of patients with hyperlipidemia. The experiments are carried out comprehensively and the result is clear and could be useful guidance for exosome research. It can be published in life after minor revisions. 

1.       In part 3.2, how the proportion was calculated needs to be described in the manuscript.

2.  Figure 3A, the label of UC, PEG, and SEC is needed to clarify each figure.

Reviewer 3 Report

The paper entitled "Comparison of Different Isolation Methods for Plasma-derived Exosomes in Patients with Hyperlipidemia" by Zhen et al. is overall well-written and results are presented in a quite clear way, even if I think the authors should add some information that are lacking.

These are my suggestions and comments for authors:

1) On line 41, citations number 4-8 are related to the use of exosomes in diagnosis and therapy as disease biomarkers and targeted drug carriers, but these references only relate to cardiovascular diseases. Since the sentence is more general, I suggest the authors to add some other references, like: 

- Antimisiaris, S.G.; Mourtas, S.; Marazioti, A. Exosomes and exosome-inspired vesicles for targeted drug delivery. Pharmaceutics 2018, 10, 218.

- Johnsen, K.B.; Gudbergsson, J.M.; Skov,M.N.; Pilgaard, L.; Moos, T.; Duroux, M.Acomprehensive overview of exosomes as drug delivery vehicles—Endogenous nanocarriers for targeted cancer therapy. Biochim. Biophys. Acta Rev. Cancer 2014, 1846, 75–87

- Huang, T.; Deng, C.X. Current progresses of exosomes as cancer diagnostic and prognostic biomarkers. Int. J. Biol. Sci. 2019, 15, 1–11.

- Ailuno, G.; Baldassari, S.; Lai, F.; Florio, T.; Caviglioli, G. Exosomes and Extracellular Vesicles as Emerging Theranostic Platforms in Cancer Research. Cells 2020, 9, 2569.

2) On line 45, the word “reproducibility” should be replaced with “reproducible”.

3) Table 1: I think that the title of the right column (“Hyperlipidemia n=12”) is not appropriate, since this column collects the values of different parameters. I think this column should not have any title, since the values reported have different units of measurements (%, mmol/L, etc.).

Moreover, in the table I suggest the authors to replace “Male sex (%)” with “Sex (%)”, and to indicate both the percentages of male and female sex.

4) The subsection “2.2. Cell culture and exosome incorporation” should be moved after the subsection “2.3. Exosome isolation and identification”. Indeed, I think that since the isolation of the vesicles temporarily precedes their labeling and internalization tests, the description of the isolation process should be placed before the internalization tests description.

Moreover, I suggest the authors to replace the term “incorporation” with “internalization” throughout the paper.

5) On line 106, the PKH67-labeled exosomes appear, but a description of the exosome labeling is lacking. I think the authors must describe the process they used to fluorescently label their vesicles.

6) On lines 108-109 the authors described the incubation of exosomes with cells; the temperature of incubation is lacking; the authors must declare it.

7) On line 123, the “EP tube” must be replaced with “Eppendorf tube” (if that is what the authors meant, it is not clear).

8) On lines 126-127, the sentence “The pellets were resuspended with half of the plasma volume of PBS and centrifuged at 12,000×g for 2 min at 4°C.” is not clear. I do not understand if the pellets were resuspended with 0.5 mL of PBS; I suggest the authors to improve the language of this sentence and to specify the added volume.

9) Lines 133-134: I think the authors should explain how they assume that fractions 4-8 are enriched with exosomes; is it based on a visual inspection, or on other kinds of analysis?

10) In the subsection “2.4. Silver staining”, the authors should add a description of the gel preparation. Moreover, it is not clear at which step the exosomes were loaded onto the gel. I suppose they are added with the developing buffer but the authors should be more clear about that. I suggest to specify the exosome loading on the gel and also to specify the amount (at least the volume) of exosomes used in this test.

11) In the subsection 3.1, the authors stated that “the exosomes isolated by UC had a larger diameter, intact structure, and less contamination” (lines 194-195). However, the UC-exosomes present a diameter of 253.02 ± 94.15 nm, which is not included in the typical exosomes diameter range that is 30-150 nm, as also stated by the authors in the introduction. Indeed, in the following paragraphs, the authors declare that with the UC method they isolate, together with exosomes, other kinds of vesicles. Therefore I wonder if it is correct to affirm that the UC-exosomes are less contaminated, since this technique led to the isolation of a heterogeneous population of vesicles.

12) On lines 248-249, the authors stated that “The concentrations of the exosomes isolated by PEG precipitation and SEC were similar, and both higher than those isolated by UC”, but few lines before they declared that the exosomes isolated by PEG-precipitation where enriched in HDL, so the high concentration obtained for the exosomes isolated by PEG precipitation is probably affected by the presence of HDL. I think the authors should comment this point.

13) Figures 4A, 4B and 4C are too small, and it is not possible to read the axis. The authors must increase the size and quality of these figures.

14) Finally, I suggest the authors to introduce, in the beginning of the Discussion section, a mention to the difficulties of isolating homogeneous vesicles populations from cell cultures; indeed, the authors should bear in mind that the definition of exosomes itself is not univocal among scientists (see MISEV2018 guidelines, doi: 10.1080/20013078.2018.1535750).

Reviewer 4 Report

This study is well conducted and written well in English, also scientifically. The content is beneficial for potential readers. Before publication, however, there are several points to be considered for revision as shown below:

Major point:

Line 28-29, 223: Although authors conclude that the exosomes isolated by SEC showed a high yield and low contamination, based on the results, the samples prepared by SEC contain much higher IgM than those by PEG and UC. In addition, it seems that content of exosome marker proteins does not differ between SEC and UC. Further consideration for conclusions is required for revision of conclusions.

Line 192-203: I think criteria for the judgment of exosome should be briefly provided here for readers. In addition, although authors mention that exosome has a typical bilayer membrane structure, the structure cannot be recognized from the PEG and SEC samples in figure 2A. Please replace them with better ones.

Line 262: P>0.10 ? (authors mention no difference between isolation methods)

Fig. 3A: Need to provide label for the panels and explanations for each panel. 

References format should be corrected: reference number in the list is redundant for each of reference.

Round 2

Reviewer 4 Report

The authors appropriately responded to what the reviewer suggested.